# Datasets for Online Controlled Experiments

**C. H. Bryan Liu** [*†]     **Ângelo Cardoso** [†]     **Paul Couturier** [*]     **Emma J. McCoy** [*]

[*] Imperial College London & [†] ASOS.com, UK

bryan.liu12@imperial.ac.uk

## Abstract

Online Controlled Experiments (OCE) are the gold standard to measure impact and guide decisions for digital products and services. Despite many methodological advances in this area, the scarcity of public datasets and the lack of a systematic review and categorization hinder its development. We present the first survey and taxonomy for OCE datasets, which highlight the lack of a public dataset to support the design and running of experiments with adaptive stopping, an increasingly popular approach to enable quickly deploying improvements or rolling back degrading changes. We release the first such dataset, containing daily checkpoints of decision metrics from multiple, real experiments run on a global e-commerce platform. The dataset design is guided by a broader discussion on data requirements for common statistical tests used in digital experimentation. We demonstrate how to use the dataset in the adaptive stopping scenario using sequential and Bayesian hypothesis tests and learn the relevant parameters for each approach.

## 1   Introduction

Online controlled experiments (OCEs) have become popular among digital technology organizations in measuring the impact of their products and services, and guiding business decisions [50, 53, 76]. Large tech companies including Google [41], Linkedin [85], and Microsoft [49] reported running thousands of experiments on any given day, and there are multiple companies established solely to manage OCEs for other businesses [11, 45]. It is also considered a key step in the machine learning development lifecycle [9, 90].

OCEs are essentially randomized controlled trials run on the Web. The simplest example, commonly known as an A/B test, splits a group of entities (e.g. users to a website) randomly into two groups, where one group is exposed to some treatment (e.g. showing a "free delivery" banner on the website) while the other act as the control (e.g. seeing the original website without any mention of free delivery). We calculate the decision metric(s) (e.g. proportion of users who bought something) based on responses from both groups, and compare the metrics using a statistical test to draw causal statements about the treatment.

The ability to run experiments on the Web allows one to interact with a large number of subjects within a short time frame and collect a large number of responses. This, together with the scale of experimentation carried out by tech organizations, should lead to a wealth of datasets describing the result of an experiment. However, there are not many publicly available OCE datasets, and we believe they were never systematically reviewed nor categorized. This is in contrast to the machine learning field, which also enjoyed its application boom in the past decade yet already has established archives and detailed categorizations for its datasets [28, 78].

We argue the lack of relevant datasets arising from real experiments hinders the further development of OCE methods (e.g. new statistical tests, bias correction, and variance reduction methods). Many statistical tests proposed relied on simulated data that impose restrictive distributional assumptions and thus may not be representative of the real-world scenario. Moreover, it may be difficult to understand how methods differ from each other and assess their relative strengths and weaknesses without a common dataset to compare them on.

35th Conference on Neural Information Processing Systems (NeurIPS 2021) Track on Datasets and Benchmarks.

To address this problem, we present the first ever survey and taxonomy for OCE datasets. Our survey identified 13 datasets, including standalone experiment archives, accompanying datasets from scholarly works, and demo datasets from online courses on the design and analysis of experiments. We also categorize these datasets based on dimensions such as the number of experiments each dataset contains, how granular each data point is time-wise and subject-wise, and whether it includes results from real experiment(s).

The taxonomy enables us to engage in a discussion on the data requirements for an experiment by systematically mapping out which data dimension is required for which statistical test and/or learning the hyperparameter(s) associated with the test. We also recognize that in practice data are often used for purposes beyond what it is originally collected for [47]. Hence, we posit the mapping is equally useful in allowing one to understand the options they have when choosing statistical tests given the format of data they possess. Together with the survey, the taxonomy helps us to identify what types of datasets are required for commonly used statistical tests, yet are missing from the public domain.

One of the gaps the survey and taxonomy identify is datasets that can support the design and running of experiments with *adaptive stopping* (a.k.a. continuous monitoring / optional stopping). We motivate their use below. Traditionally, experimenters analyze experiments using Null Hypothesis Statistical Tests (NHST, e.g. a Student's $t$-test). These tests require one to calculate and commit to a required sample size based on some expected treatment effect size, all prior to starting the experiment. Making extra decisions during the experiment, be it stopping the experiment early due to seeing favorable results, or extending the experiment as it teeters "on the edge of statistical significance" [61], is discouraged as they risk one having more false discoveries than intended [35, 60].

Clearly, the restrictions above are incompatible with modern decision-making processes. Businesses operating online are incentivized to deploy any beneficial and roll back any damaging changes as quickly as possible. Using the "free delivery" banner example above, the business may have calculated that they require four weeks to observe enough users based on an expected 1% change in the decision metric. If the experiment shows, two weeks in, that the banner is leading to a 2% improvement, it will be unwise not to deploy the banner to all users simply due to the need to run the experiment for another two weeks. Likewise, if the banner is shown leading to a 2% loss, it makes every sense to immediately terminate the experiment and roll back the banner to stem further losses.

As a result, more experimenters are moving away from NHST and adopting adaptive stopping techniques. Experiments with adaptive stopping allow one to decide when to stop an experiment (i.e. stopping it earlier or prolonging it) based on the sample responses observed so far without compromising the statistical validity of false positive/discovery rate control. To encourage further development in this area, both in methods and data, we release the ASOS Digital Experiments Dataset, which contains daily checkpoints of decision metrics from multiple, real OCEs run on the global online fashion retail platform.

The dataset design is guided by the requirements identified by the mapping between the taxonomy and statistical tests, and to the best of our knowledge, is the first public dataset that can support the end-to-end design and running of online experiments with adaptive stopping. We demonstrate it can indeed do so by (1) running a sequential test and a Bayesian hypothesis test on all the experiments in the dataset, and (2) estimating the value of hyperparameters associated with the tests. While the notion of ground-truth does not exist in real OCEs, we show the dataset can also act as a quasi-benchmark for statistical tests by comparing results from the tests above with that of a $t$-test.

To summarize, our contributions are:

1. (Sections 2 & 3) We create, to the best of our knowledge, the first ever taxonomy on online controlled experiment datasets and apply it to publicly available datasets;

2. (Section 4) We map the relationship between the taxonomy to statistical tests commonly used in experiments by identifying the minimally sufficient set of statistics and dimensions required in each test. The mapping, which also applies to offline and non-randomized controlled experiments, enables experimenters to quickly identify the data collection requirements for their experiment design (and conversely the test options available given the data availability); and

3. (Section 5) We make available, to the best of our knowledge, the first real, multi-experiment time series dataset, enabling the design and running of experimentation with adaptive stopping.[1]

---

[1]Link to the dataset and accompanying datasheet: `https://osf.io/64jsb/`

## 2 A Taxonomy for Online Controlled Experiment Datasets

We begin by presenting a taxonomy on OCE datasets, which is necessary to characterize and understand the results of a survey. To the best of our knowledge, there are no surveys nor taxonomies specifically on this topic prior to this work. While there is a large volume of work concerning the categorization of datasets in machine learning [28, 78], of research work in the online randomized controlled experiment methods [3, 4, 32, 69], and of general experiment design [43, 52], our search on Google Scholar and Semantic Scholar using combinations of the keywords "online controlled experiment"/"A/B test", "dataset", and "taxonomy"/"categorization" yields no relevant results.

The taxonomy focuses on the following four main dimensions:

**Experiment count**   A dataset can contain the data collected from a *single* experiment or *multiple* experiments. Results from a single experiment are useful for demonstrating how a test works, though any learning should ideally involve multiple experiments. Two closely related but relatively minor dimensions are the *variant count* (number of control/treatment groups in the experiment) and the *metric count* (number of performance metrics the experiment is tracking). Having an experiment with multiple variants and metrics enables the demonstration of methods such as false discovery rate control procedures [8] and learning the correlation structure within an experiment.

**Response granularity**   Depending on the experiment analysis requirements and constraints imposed by the online experimentation platform, the dataset may contain data aggregated to various levels. Consider the "free delivery" banner example in Section 1, where the website users are randomly allocated to the treatment (showing the banner) and control (not showing the banner) groups to understand whether the banner changes the proportion of users who bought something. In this case, each individual user is considered a randomization unit [50].

A dataset may contain, for each experiment, only summary statistics on the *group* level, e.g. the proportion of users who have bought something in the control and treatment groups respectively. It can also record one response per *randomization unit*, with each row containing the user ID and whether the user bought something. The more detailed activity logs at a *sub-randomization unit* level can also have each row containing information about a particular page view from a particular user.

**Time granularity**   An experiment can last anytime between a week to many months [50], which provides many possibilities in recording the result. A dataset can opt to record the *overall result only*, showing the end state of an experiment. It may or may not come with a *timestamp* for each randomization unit or experiment if there are multiple instances of them. It can also record *intermediate checkpoints* for the decision metrics, ideally at regular intervals such as *daily* or *hourly*. These checkpoints can either be a *snapshot* of the interval (recording activities between time $t$ and $t + 1$, time $t + 1$ and $t + 2$, etc.) or *cumulative* from the start of the experiment (recording activities between time 0 and 1, time 0 and 2, etc.).

**Syntheticity**   A dataset can record data generated from a *real* process. It can also be *synthetic*—generated via simulations with distributional assumptions applied. A dataset can also be *semi-synthetic* if it is generated from a real-life process and subsequently augmented with synthetic data.

Note we can also describe datasets arising from any experiments (including offline and non-randomized controlled experiments) using these four dimensions. We will discuss in Section 4 how these dimensions map to common statistical tests used in online experimentation.

In addition, we also record the *application domain*, *target demographics*, and the *temporal coverage* of the experiment(s) featured in a dataset. In an age when data are often reused, it is crucial for one to understand the underlying context, and that learnings from a dataset created under a certain context may not translate to another context. We also see the surfacing of such context as a way to promote considerations in fairness and transparency for experimenters as more experiment datasets become available [44]. For example, having target demographics information on a meta-level helps experimenters to identify who were involved, or perhaps more importantly, who were *not* involved in experiments and could be adversely impacted by a treatment that is effectively untested.

Finally, two datasets can also differ in their medium of *documentation* and the presence/absence of a *data management / long-term preservation plan*. The latter includes the hosting location, the presence/absence of a DOI, and the type of license. We record these attributes for the datasets surveyed below for completeness.

# 3    Public Online Controlled Experiment Datasets

Here we discuss our approach to produce the first ever survey on OCE datasets and present its results. The survey is compiled via two search directions, which we describe below. For both directions, we conduct a first round search in May 2021, with follow up rounds in August and October 2021 to ensure we have the most updated results.

We first search on the vanilla Google search engine using the keywords "Online controlled experiment "dataset"", "A/B test "dataset"", and "Multivariate test "dataset"". For each keyword, we inspect the first 10 pages of the search result (top 100 results) for scholarly articles, web pages, blog posts, and documents that may host and/or describe a publicly available OCE dataset. The search term "dataset" is in double quotes to limit the search results to those with explicit mention of dataset(s). We also search on specialist data search engines/hosts, namely on Google Dataset Search (GDS) and Kaggle, using the keywords "Online controlled experiment(s)" and "A/B test(s)". We inspect the metadata and description for all the results returned (except for GDS, where we inspect the first 100 results for "A/B test(s)") for relevant datasets as defined below.[2]

A dataset must record the result arising from a randomized controlled experiment run online to be included in the survey. The criterion excludes experimental data collected from offline experiments, e.g. those in agriculture [84], medicine [27], and economics [29]. It also excludes datasets used to perform quasi-experiments and observational studies, e.g. the LaLonde dataset used in econometrics [21] and datasets constructed for uplift modeling tasks [25, 40].[3]

The result is presented in Table 1. We place the 13 OCE datasets identified in this exercise along the four taxonomy dimensions defined in Section 2 and record the additional features. These datasets include two standalone archives for online media and education experiments respectively [56, 71], plus two accompanying datasets for peer-reviewed research articles [75, 87]. There are also tens of Kaggle datasets, blog posts, and code repositories that describe and/or duplicate one of the five example datasets used in five different massive open online courses on online controlled experiment design and analysis [10, 12, 36, 37, 77]. Finally, we identify three standalone datasets hosted on Kaggle with relatively light documentation [5, 31, 48].

From the table, we observe a number of gaps in OCE dataset availability, the most obvious one being the lack of datasets that record responses at a sub-randomization unit level. In the sections below, we will identify more of these gaps and discuss their implications for OCE analysis.

# 4    Matching Dataset Taxonomy with Statistical Tests

Specifying the data requirements (or structure) and performing statistical tests are perhaps two of the most common tasks carried out by data scientists. However, the link between the two processes is seldom mapped out explicitly. It is all too common to consider from scratch the question "I need to run this statistical test, how should I format my dataset?" (or more controversially, "I have this dataset, what statistical tests can I run?" [47]) for every new project/application, despite the list of possible dataset dimensions and statistical tests remaining largely the same.

We aim to speed up the process above by describing what summary statistics are required to perform common statistical tests in OCE, and link the statistics back to the taxonomy dimensions defined in Section 2. The exercise is similar to identifying the sufficient statistic(s) for a statistical model [33], though the identification is done for the encapsulating statistical inference procedure, with a practical focus on data dimension requirements. We do so by stating the formula used to calculate the corresponding effect sizes and test statistics and observe the summary statistics required in common. The general approach enables one to also apply the resultant mapping to any experiments that involve

---

[2]Searching for the keyword "Online controlled experiment" on GDS and Kaggle returned 42 and 7 results respectively, and that for "A/B test" returned "100+" and 286 results respectively. Curiously, replacing "experiment" and "test" in the keywords with their plural form changes the number of results, with the former returning 6 and 10 results on GDS and Kaggle respectively, and the latter returning "100+" and 303 results respectively.

[3]Uplift modeling (UM) tasks for online applications often start with an OCE [67] and thus we can consider UM datasets as OCE datasets with extra randomization unit level features. The nature of the tasks is different though: OCEs concern validating the average treatment effect across the population using a statistical test, whereas UM concerns modeling the conditional average treatment effect for each individual user, making it more a general causal inference task that is outside the scope of this survey.

| Dataset Name | Ref. | Experiment Count - Variant / Metric Count | Response Granularity | Time Granularity | Syntheticity | Application Domain | Target Demographic | Temporal Coverage | Documentation | Data management / long-term preservation plan |
|---|---|---|---|---|---|---|---|---|---|---|
| Upworthy Research Archive | [56] | Multiple (32,487) - 2-14 / 2 (C) | Group | Overall result only Timestamp per expt. | Real | Media & Ads Copy/Creative | Mostly English-speaking users in USA | Jan 2013 – Apr 2015 | Peer reviewed data article | Host: Open Science Framework DOI: ✓ (see [56]) Licence: CC BY 4.0 |
| ASSISTments Dataset from Multiple Randomized Controlled Experiments | [71] | Multiple (22) - 2 / 2 (BC) | Rand. Unit | Overall result only ✗ timestamp | Real | Education Teaching model | Mostly middle school students (age 11-14) in/near MA, USA | 2013 – 2015 | Peer reviewed data article | Host: Author website DOI: Unknown Licence: Unknown |
| A/B Testing Web Analytics Data (From [88]) | [87] | Single - 5 / 2 (C) | Group | Overall result only ✓ timestamp | Real | Education UX/UI change | Mostly English-speaking university library users | May 2013 – Jun 2013 | Accompanying dataset to peer-reviewed research article | Host: University library DOI: ✓ (see [87]) Licence: CC BY-SA 4.0 |
| Dataset of two experiments of the application of gamified peer assessment model into online learning environment MeuTutor (From [74]) | [75] | Multiple (2) - 3+2 / 3+3 (R+C) | Rand. Unit | Overall result only ✗ timestamp | Real | Education Teaching model | High school students in Brazil taking ENEM (age 17) | Jul 2015 – Aug 2015 | Peer reviewed data article | Host: Journal website DOI: ✓ (see [75]) Licence: CC BY 4.0 |
| Udacity Free Trial Screener Experiment (From Udacity A/B Testing Course - Final Project [36]) | See e.g. [62, 72, 80] | Single - 2 / 4 (C) | Group | Daily checkpoint Snapshot | Real | Education UX/UI change | Mostly English-speaking users | Unknown | Blog posts & Kaggle notebooks e.g. [57, 70, 89] | Host: Kaggle / GitHub (multiple) DOI: Unknown Licence: Unknown |
| "Analyse A/B Test Results" Dataset (From Udacity Online Data Analyst Course - Project 3 [77]) | See e.g. [2, 20, 65] | Single - 2 / 1 (B) | Rand. Unit | Overall result only Timestamp per RU | Unknown | E-commerce UX/UI change | Unknown | Unknown | Blog posts & Kaggle notebooks e.g. [16, 54, 68] | Host: Kaggle / GitHub (multiple) DOI: Unknown Licence: Unknown |
| Mobile Games A/B Testing with Cookie Cats (DataCamp project [10]) | See e.g. [30, 73, 86] | Single - 2 / 3 (BC) | Rand. Unit | Overall result only ✗ timestamp | Real | Gaming Design change | Unknown (likely Facebook users) | Unknown | Kaggle notebook | Host: Kaggle / Course website DOI: Unknown Licence: Unknown |
| Experiment Dataset (From DataCamp A/B Testing in R Course [12]) | [13] | Single - 2 / 1 (B) | Rand. Unit | Overall result only Timestamp per RU | Synthetic | Tech UX/UI Change | N/A | N/A | Course notes Blog posts | Host: Course website DOI: Unknown Licence: Unknown |
| Data Visualization Website - April 2018 (From DataCamp A/B Testing in R Course [12]) | [14] | Single - 2 / 4 (BR) | Rand. Unit | Overall result only Timestamp per RU | Synthetic | Tech UX/UI Change | N/A | N/A | Course notes | Host: Course website DOI: Unknown Licence: Unknown |
| AB Testing Result (From Customer Analytics and A/B Testing in Python [37]) | [38] | Single - 2 / 2 (CR) | Rand. Unit | Overall result only Timestamp per RU | Unknown | Tech UX/UI Change | Unknown | Jan 2014 – Jan 2018 | Course notes | Host: Course website DOI: Unknown Licence: Unknown |
| Grocery Website Data for AB Test | [48] | Single - 2 / 1 (B) | Rand. Unit | Overall result only ✗ timestamp | Unknown | E-commerce UX/UI change | Unknown | Unknown | Kaggle notebook | Host: Kaggle DOI: Unknown Licence: Unknown |
| Ad A/B Testing (aka SmartAd AB Data) | [31] | Single - 2 / 2 (B) | Rand. Unit | Overall result only Timestamp per RU | Unknown | Media & Ads Display ads | Unknown | Jul 2020 | Kaggle notebook | Host: Kaggle DOI: N/A Licence: CC BY-SA 3.0 |
| Synthetical A/B-Tests | [5] | Multiple (25,856) - 2 / 1 (R) | Group. | Overall result only ✗ timestamp | Synthetic | N/A | N/A | N/A | Kaggle notebook | Host: Kaggle DOI: Unknown Licence: CDLA-Sharing-1.0 |

Table 1: Results from the first ever survey of OCE datasets. The 13 datasets identified are placed on the four taxonomy dimensions defined in Section 2 together with the additional attributes recorded. In the Experiment Count column, a second-line value (corresponding to Variant/Metric Count) of "$x$ / $y$ (BCLR)" means the dataset features $x$ variants and $y$ metrics, with the metrics based on **B**inary, **C**ount, **L**ikert-scale, and **R**eal-valued responses. In the Response Granularity and Time Granlarity columns, Randomization Unit is abbreviated as RU or Rand. Unit. Note that a large proportion of resources are accessed via links to non-scholarly articles—blog plots, Kaggle dataset pages, and GitHub repositories. These resources may not persist over time.

a two-sample statistical test, including offline experiments and experiments without a randomized control. For brevity, we will refrain from discussing the full model assumptions as well as their applicability. Instead, we point readers to the relevant work in the literature.

## 4.1 Effect size and Welch's $t$-test

We consider a two-sample setting and let $X_1, \cdots, X_N$ and $Y_1, \cdots, Y_M$ be i.i.d. samples from the distributions $F_X(\cdot)$ and $F_Y(\cdot)$ respectively. We assume the first two moments exist for the distributions $F_X$ and $F_Y$, with their mean and variance denoted $(\mu_X, \sigma_X^2)$ and $(\mu_Y, \sigma_Y^2)$ respectively. We also denote the sample mean and variance of the two samples $(\bar{X}, s_X^2)$ and $(\bar{Y}, s_Y^2)$ respectively.

Often we are interested in the difference between the mean of the two distributions $\Delta = \mu_Y - \mu_X$, commonly known as the *effect size* (of the difference in mean) or the *average treatment effect*. A standardized effect size enables us to compare the difference across many experiments and is thus useful in meta-analyses. One commonly used effect size is Cohen's $d$, defined as the difference in sample means divided by the pooled sample standard deviation [18]:

$$d = \left(\bar{Y} - \bar{X}\right) \Big/ \sqrt{\frac{(N-1)s_X^2 + (M-1)s_Y^2}{N + M - 2}} \, . \tag{1}$$

We are also interested in whether the samples carry sufficient evidence to indicate $\Delta$ is different from a prescribed value $\theta_0$. This can be done via a hypothesis test with $H_0 : \Delta = \theta_0$ and $H_1 : \Delta \neq \theta_0$.[4] One of the most common statistical test used in online controlled experiments is the Welch's $t$-test [82], in which we calculate the test statistic $t$ as follow:

$$t = \left(\bar{Y} - \bar{X}\right) \Big/ \sqrt{\frac{s_X^2}{N} + \frac{s_Y^2}{M}} \, . \tag{2}$$

We observe that in order to calculate the two stated quantities above, we require six quantities: two means $(\bar{X}, \bar{Y})$, two (sample) variances $(s_X^2, s_Y^2)$, and two counts $(N, M)$. We call these quantities *Dimension Zero (D0)* quantities as they are the bare minimum required to run a statistical test—these quantities will be expanded along the taxonomy dimensions defined in Section 2.

**Cluster randomization / dependent data**    The sample variance estimates $(s_X^2, s_Y^2)$ may be biased in the case where cluster randomization is involved. Using again the "free delivery" banner example, instead of randomly assigning each individual user to the control and treatment groups, the business may randomly assign postcodes to the two groups, with all users from the same postcode getting the same version of the website. In this case, user responses may become correlated, which violates the independence assumptions in statistical tests. Common workarounds including the use of bootstrap [7] and the Delta method [24] generally require access to *sub-randomization unit* responses.

## 4.2 Experiments with adaptive stopping

As discussed in Section 1, experiments with adaptive stopping are getting increasingly popular among the OCE community. Here we motivate the data requirement for statistical tests in this domain by looking at the quantities required to calculate the test statistics for a Mixture Sequential Probability Ratio Test (mSPRT) [45] and a Bayesian hypothesis test using Bayes factor [23], two popular approaches in online experimentation. There are many other tests that support adaptive stopping [59, 79], though the data requirements, in terms of the dimensions defined in Section 2, should be largely identical.

We first observe running a mSPRT with a normal mixing distribution $H = \mathcal{N}(\theta_0, \tau^2)$ involves calculating the following test statistic upon observing the first $n$ $X_i$ and $Y_j$ (see Eq. (11) in [45]):

$$\tilde{\Lambda}_n^{H, \theta_0} = \sqrt{\frac{\sigma_X^2 + \sigma_Y^2}{\sigma_X^2 + \sigma_Y^2 + n\tau^2}} \exp\left(\frac{n^2 \tau^2 (\bar{Y}_n - \bar{X}_n - \theta_0)^2}{2(\sigma_X^2 + \sigma_Y^2)(\sigma_X^2 + \sigma_Y^2 + n\tau^2)}\right), \tag{3}$$

where $\bar{X}_n = n^{-1} \sum_{i=1}^{n} X_i$ and $\bar{Y}_n = n^{-1} \sum_{j=1}^{n} Y_j$ represent the sample mean of the $X$s and $Y$s up to sample $n$ respectively, and $\tau^2$ is a hyperparameter to be specified or learned from data.

---

[4]There are other ways to specify the hypotheses such as that in superiority and non-inferiority tests [19], though they are unlikely to change the data requirement as long as it remains anchored on $\theta_0$.

For Bayesian hypothesis tests using Bayes factor, we calculate the (square root of) Wald test statistic upon seeing the first $n$ $X_i$ and first $m$ $Y_j$ [22, 23]:

$$W_{n,m} = \frac{\bar{Y}_m - \bar{X}_n}{\sqrt{\left(\frac{\sigma_X^2}{n} + \frac{\sigma_Y^2}{m}\right)}} = \underbrace{\frac{\bar{Y}_m - \bar{X}_n}{\sqrt{\left(\frac{\sigma_X^2}{n} + \frac{\sigma_Y^2}{m}\right)/\left(\frac{1}{n} + \frac{1}{m}\right)}}}_{\delta_{n,m}} \underbrace{\frac{1}{\sqrt{\frac{1}{n} + \frac{1}{m}}}}_{\sqrt{E_{n,m}}} \, , \tag{4}$$

where $\delta_{n,m}$ and $E_{n,m}$ are the *effect size* (standardized by the pooled variance) and the *effective sample size* of the test respectively. In OCE it is common to appeal to the central limit theorem and assume a normal likelihood for the effect size, i.e. $\delta_{n,m} \sim \mathcal{N}(\mu, 1/E_{n,m})$. We then compare the hypotheses $H_0 : \mu = \theta_0$ and $H_1 : \mu \sim \mathcal{N}(\theta_0, V^2)$ by calculating the Bayes factor [46]:

$$BF_{n,m} = \frac{f(\delta_{n,m}|H_1)}{f(\delta_{n,m}|H_0)} = \frac{\phi(\delta_{n,m}; \theta_0, V^2 + 1/E_{n,m})}{\phi(\delta_{n,m}; \theta_0, 1/E_{n,m})} \, , \tag{5}$$

where $\phi(\cdot\,; a, b)$ is the PDF of a normal distribution with mean $a$ and variance $b$, and $V^2$ is a hyperparameter that we specify or learn from data.

During an experiment with adaptive stopping, we calculate the test statistics stated above many times for different $n$ and $m$. This means a dataset can only support the running of such experiments if it contains *intermediate checkpoints* for the counts $(n, m)$ and the means $(\bar{X}_n, \bar{Y}_m)$, ideally *cumulative* from the start of the experiment. Often one also requires the variances at the same time points (see below). The only exception to the dimensional requirement above is the case where the dataset contains responses at a *randomization unit* or finer level of granularity, and despite recording the *overall results only*, has a *timestamp per randomization unit*. Under this special case, we will still be able to construct the cumulative means $(\bar{X}_n, \bar{Y}_m)$ for all relevant values of $n$ and $m$ by ordering the randomization units by their associated timestamps.

**Learning the effect size distribution (hyper)parameters**   The two tests introduced above feature some hyperparameters ($\tau^2$ and $V^2$) that have to be specified or learned from data. These parameters characterize the prior belief of the effect size distribution, which will be the most effective if it "matches the distribution of true effects across the experiments a user runs" [45]. Common parameter estimation procedures [1, 6, 39] require results from *multiple related experiments*.

**Estimating the response variance**   In the equations above, the response variance of the two samples $\sigma_X^2$ and $\sigma_Y^2$ are assumed to be known. In practice we often use the plug-in empirical estimates $(s_X^2)_n$ and $(s_Y^2)_m$—the sample variances for the first $n$ $X_i$ and first $m$ $Y_j$ respectively, and thus the data dimensional requirement is identical to that of the counts and means as discussed above. In the case where the plug-in estimate may be biased due to dependent data, we will also require a *sub-randomization unit* response granularity (see Section 4.1).

## 4.3   Non-parametric tests

We also briefly discuss the data requirements for non-parametric tests, where we do not impose any distributional assumptions on the responses but compare the hypotheses $H_0 : F_X \equiv F_Y$ and $H_1 : F_X \neq F_Y$, where we recall $F_X$ and $F_Y$ are the distributions of the two samples.

One of the most commonly used (frequentist) non-parametric tests in OCE, the Mann-Whitney $U$-test [55], calculates the following test statistic:

$$U = \sum_{i=1}^{N} \sum_{j=1}^{M} S(X_i, Y_j), \quad \text{where } S(X, Y) = \begin{cases} 1 & \text{if } Y < X, \\ 1/2 & \text{if } Y = X, \\ 0 & \text{if } Y > X. \end{cases} \tag{6}$$

While a rank-based method is available for large $N$ and $M$, both methods require the knowledge of all the $X_i$ and $Y_j$. Such requirement is the same for other non-parametric tests, e.g. the Wilcoxon signed-rank [83], Kruskal-Wallis [51], and Kolmogorov–Smirnov tests [26]. This suggests a dataset can only support a non-parametric test if it at least provides responses at a *randomization unit* level.

We conclude by showing how we can combine the individual data requirements above to obtain the requirement to design and/or run experiments for more complicated statistical tests. This is possible due to the orthogonal design of the taxonomy dimensions. Consider an experiment with adaptive

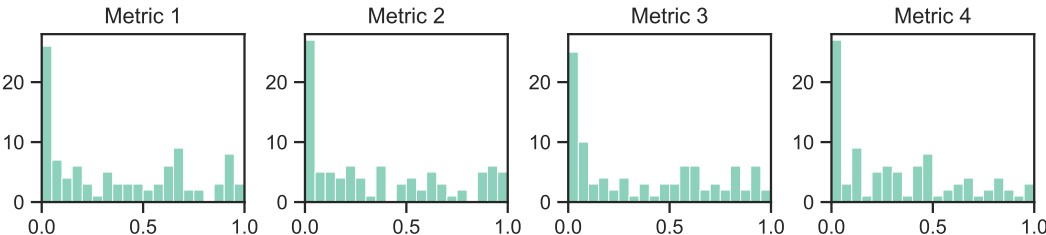

Figure 1: Distribution of $p$-values attained by the 99 OCEs in the ASOS Digital Experiment Dataset using Welch's $t$-tests, split by decision metrics. The leftmost bar in each histogram represents experiments with $p < 0.05$. Here we treat OCEs with multiple variants as multiple independent OCEs.

stopping using Bayesian non-parametric tests (e.g. with a Pólya Tree prior [17, 42]). It involves a non-parametric test and hence requires responses at a *randomization unit* level. It computes multiple Bayes factors for adaptive stopping and hence requires *intermediate checkpoints* for the responses (or a timestamp for each randomization unit). Finally, to learn the hyperparameters of the Pólya Tree prior it requires *multiple related experiments*. The substantial data requirement along 3+ dimensions perhaps explains the lack of relevant OCE datasets and the tendency for experimenters to use simpler statistical tests for the day-to-day design and/or running of OCEs.

## 5 A Novel Dataset for Experiments with Adaptive Stopping

We finally introduce the ASOS Digital Experiments Dataset, which we believe is the first public dataset that supports the end-to-end design and running of OCEs with adaptive stopping. We motivate why this is the case, provide a light description of the dataset (and a link to the more detailed accompanying datasheet),[5] and showcase the capabilities of the dataset via a series of experiments. We also discuss the ethical implications of releasing this dataset.

Recall from Section 4.2 that in order to support the end-to-end design and running of experiments with adaptive stopping, we require a dataset that (1) includes *multiple* related experiments; (2) is *real*, so that any parameters learned are reflective of the real-world scenario; and either (3a) contains *intermediate checkpoints* for the summary statistics during each experiment (i.e. time-granular), or (3b) contains responses at a *randomization unit* granularity with a timestamp for each randomization unit (i.e. response-granular with timestamps).

None of the datasets surveyed in Section 3 meet all three criteria. While the Upworthy [56], ASSISTments [71], and MeuTutor [75] datasets meet the first two criteria, they all fail to meet the third.[5] The Udacity Free Trial Screener Experiment dataset meets the last two criteria by having results from a real experiment with daily snapshots of the decision metrics (and hence time-granular), which supports the running of an experiment with adaptive stopping. However, the dataset only contains a single experiment, which is not helpful for learning the effect size distribution (the design).

The ASOS Digital Experiments Dataset contains results from OCEs run by a business unit within ASOS.com, a global online fashion retail platform. In terms of the taxonomy defined in Section 2, the dataset contains *multiple* (78), *real* experiments, with two to five variants in each experiment and four decision metrics based on binary, count, and real-valued responses. The results are aggregated on a *group* level, with *daily or 12-hourly checkpoints* of the metric values *cumulative* from the start of the experiment. The dataset design meets all the three criteria stated above and hence differentiates itself from other public datasets.

We provide readers with an accompanying datasheet (based on [34]) that provides further information about the dataset. We also host the dataset on Open Science Framework to ensure it is easily discoverable and can be preserved long-term.[1] It is worth noting that the dataset is released with the intent to support development in the statistical methods required to run OCEs. The experiment results shown in the dataset are not representative of ASOS.com's overall business operations, product development, or experimentation program operations, and no conclusion of such should be drawn from this dataset.

---

[5]All three report the overall results only and hence are not time-granular. The Upworthy dataset reports group-level statistics and hence is not response-granular. The ASSISTments and MeuTutor datasets are response-granular but they lack the timestamp to order the samples.

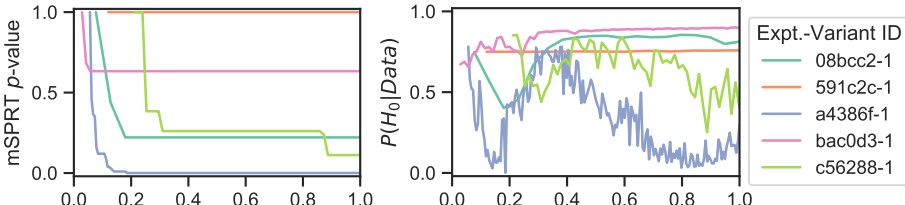

Figure 2: Change in (left) $p$-values in a mixed Sequential Probability Ratio Test ($\tau^2 = $ `5.92e-06`) and (right) posterior belief in the null hypothesis ($\pi(H_0|\text{data})$) in a Bayesian hypothesis test ($V^2 = $ `5.93e-06`, $\pi(H_0) = 0.75$) during the experiment for five experiments selected at random. The experiment duration (x-axis) is normalized by its overall runtime. Only results for Metric 4 is shown.

## 5.1 Potential use cases

**Meta-analyses** The multi-experiment nature of the dataset enables one to perform meta-analyses. A simple example is to characterize the distribution of $p$-values (under a Welch's $t$-test) across all experiments (see Figure 1). We observe there are roughly a quarter of experiments in this dataset attaining $p < 0.05$, and attribute this to the fact that what we experiment in OCEs are often guided by what domain experts think may have an impact. Having that said, we invite external validation on whether there is evidence for data dredging using e.g. [58].

**Design and running of experiments with adaptive stopping** We then demonstrate the dataset can indeed support OCEs with adaptive stopping by performing a mixed Sequential Probability Ratio test (mSPRT) and a Bayesian hypothesis test via Bayes factor for each experiment and metric. This requires learning the hyperparameters $\tau^2$ and $V^2$. We learn, for each metric, a naïve estimate for $V^2$ by collating the $\delta_{n,m}$ (see (4)) at the end of each experiment and taking their sample variance. This yields the estimates `1.30e-05`, `1.07e-05`, `6.49e-06`, and `5.93e-06` for the four metrics respectively. For $\tau^2$, we learn near-identical naïve estimates by collating the value of Cohen's $d$ (see (1)) instead. However, as $\tau^2$ captures the spread of unstandardized effect sizes, we specify in each test $\tau^2 = d \cdot (s_X^2)_n$, where $(s_X^2)_n$ is the sample variance of all responses up to the $n^{\text{th}}$ observation in that particular experiment. The Bayesian tests also require a prior belief in the null hypothesis being true ($\pi(H_0)$)—we set it to $0.75$ based on what we observed in the $t$-tests above.

We then calculate the $p$-value in mSPRT and the posterior belief in the null hypothesis ($\pi(H_0|\text{data})$) in the Bayesian test for each experiment and metric at each daily/12-hourly checkpoint, following the procedures stated in [45] and [23, 46] respectively. We plot the results for five experiments selected at random in Figure 2, which shows the $p$-value for a mSPRT is monotonically non-increasing, while the posterior belief for a Bayesian test can fluctuate depending on the effect size observed so far.

**A quasi-benchmark for adaptive stopping methods** Real online controlled experiments, unlike machine learning tasks, generally do not have a notion of ground truth. The use of quasi-ground truth enables the comparison between two hyperparameter settings of the same adaptive stopping method or two adaptive stopping methods. Using as quasi-ground truth the significant / not significant verdict from a Welch's $t$-test at the end of the experiment as the "ground truth", we could then compare this "ground truth" to the significant / not significant verdict of a mSPRT at different stages of individual experiments. This yields many "confusion matrices" over different stages of individual experiments where a "Type I error" corresponds to cases where a Welch's $t$-test gives a not significant result and a mSRPT reports a significant result, a confusion matrix for the end of each experiment can be seen in Table 2. As the dataset was collected without early stopping it allows us to perform sensitivity analysis and optimization on the hyperparameters of mSPRT under what can be construed as a "precision-recall" tradeoff of statistically significant treatments.

**Other use cases** The time series nature of this dataset enables one to detect bias (of the estimator) across time, e.g. those that are caused by concept drift or feedback loops. In the context of OCEs, [15] described a number of methods to detect invalid experiments over time that may be run on this dataset. Moreover, being both a multi-experiment and time series dataset also enables one to learn the correlation structure across experiments, decision metrics, and time [66, 81].

## 5.2 Ethical considerations

We finally discuss the ethical implications of releasing the dataset, touching on data protection and anonymization, potential misuses, and the ethical considerations for running OCEs in general.

| $t$-test | Significant | | Not significant | |
| --- | --- | --- | --- | --- |
| mSPRT | Significant | Not significant | Significant | Not significant |
| Metric 1 | 19 | 7 | 2 | 71 |
| Metric 2 | 20 | 7 | 18 | 49 |
| Metric 3 | 16 | 9 | 6 | 63 |
| Metric 4 | 16 | 11 | 4 | 63 |

Table 2: Comparing the number of statistically significant / not significant results reported by a Welch's $t$-test and an mSPRT at the end of an experiment for all four metrics.

**Data protection and anonymization**   The dataset records aggregated activities of hundreds of thousands or millions of website users for business measurement purposes and hence it is impossible to identify a particular user. Moreover, to minimize the risk of disclosing business sensitive information, all experiment context is either removed or anonymized such that one should not be able to tell who is in an experiment, when is it run, what treatment does it involve, and what decision metrics are used. We refer readers to the accompanying datasheet[5] for further details in this area.

**Potential misuses**   An OCE dataset, no matter how anonymized it is, reflects the behavior of its participants under a certain application domain and time. We urge potential users of this dataset to exercise caution when attempting to generalize the learnings. It is important to emphasize that the learnings are different from the statistical methods and processes that are demonstrated on this dataset. We believe the latter are generalizable, i.e. they can be applied on other datasets with similar data dimensions regardless of the datasets' application domain, target demographics, and temporal coverage, and appeal for potential users of the dataset to focus on such.

One example of generalizing the learnings is the use of this dataset as a full performance benchmark. As discussed above, this dataset does not have a notion of ground truth and any quasi-ground truths constructed are themselves a source of bias to estimators. Thus, experiment design comparisons need to be considered at a theoretical level [52]. Another example will be directly applying the value of hyperparameter(s) obtained while training a model on this dataset to another dataset. While this may work for similar application domains, the less similar they are the less likely the hyperparameters learned will transfer. This may introduce risk in incurring bias both on the estimator and in fairness.

**Running OCEs in general**   The dataset is released with the aim to support experiments with adaptive stopping, which will enable a faster experimentation cycle. As we run more OCEs, which are ultimately human subjects research, the ethical concerns will naturally mount. We reiterate the importance of the following three principles when we design and run experiments [63, 64]: respect for persons, beneficence (properly assess and balance the risks and benefits), and justice (ensure participants are not exploited), and refer readers to Chapter 9 of [50] and its references for further discussions in this area.

## 6   Conclusion

Online controlled experiments (OCE) are a powerful tool for online organizations to assess their digital products' and services' impact. To safeguard future methodological development in the area, it is vital to have access to and have a systematic understanding of relevant datasets arising from real experiments. We described the result of the first ever survey on publicly available OCE datasets, and provided a dimensional taxonomy that links the data collection and statistical test requirements. We also released the first ever dataset that can support OCEs with adaptive stopping, which design is grounded on a theoretical discussion between the taxonomy and statistical tests. Via extensive experiments, we also showed that the dataset is capable of addressing the identified gap in the literature.

Our work on surveying, categorizing, and enriching the publicly available OCE datasets is just the beginning and we invite the community to join in the effort. As discussed above we have yet to see a dataset that can support methods dealing with correlated data due to cluster randomization, or the end-to-end design and running of experiments with adaptive stopping using Bayesian non-parametric tests. We also see ample opportunity to generalize the survey to cover datasets arising from uplift modeling tasks, quasi-experiments, and observational studies. Finally, we can further expand the taxonomy, which already supports datasets from all experiments, with extra dimensions (e.g. number of features to support stratification, control variate, and uplift modeling methods) as the area matures.

## Acknowledgments and Disclosure of Funding

CHBL is part-funded by the EPSRC CDT in Modern Statistics and Statistical Machine Learning at Imperial College London and University of Oxford (StatML.IO) and ASOS.com. The authors thank their colleagues, participants in CODE@MIT 2021, and the anonymous reviewers for suggesting many improvements to the original manuscript.

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
