# OpenReview forum: "Datasets for Online Controlled Experiments"
_NeurIPS.cc/2021/Track/Datasets_and_Benchmarks/Round2 — NeurIPS 2021 Datasets and Benchmarks Track (Round 2)_

### Official Review · Reviewer_rsHD · 2021-09-20
**Datasets for Online Controlled Experiments**

**Rating:** 6
**Confidence:** 2
**Correctness:** Not qualified to assess
**Clarity:** The paper is clear and well written.

**Strengths:**

This paper explains the importance of OCEs and early stopping, the range of experimental setups, and the difficulty of choosing amongst them.

**Weaknesses:**

The authors did not include an adequate consideration of the ethical implications of releasing this dataset, or the limitations of the study for generalizability

**Additional Feedback:**

Resolve comments made under weaknesses/ethics.

**Documentation:**

Yes

**Ethics:**

See weaknesses... this discussion, and the limits of generalizability (an important ethical concern) were not adequately discussed.

**Relation To Prior Work:**

The authors cite a significant literature gap, but position their paper well in relation to existing work.

**Summary And Contributions:**

This paper offers a dataset containing multiple online controlled experiments featuring early stopping, to assist in OCE design.

---

> ### Author Response · Authors · 2021-09-28
> **Response to Reviewer rsHD (part 1)**
>
> Thank you for raising concerns on the ethical implications and the generalizability of the study and dataset.
>
> > The authors did not include an adequate consideration of the ethical implications of releasing this dataset.
>
> We have discussed the implications of releasing the dataset in terms of subject consent and data protection in the accompanying datasheet. In particular, the dataset records aggregated activities of hundreds of thousands or millions of website users for business measurement purposes and hence it is impossible to identify a particular user. The dataset is reviewed by ASOS’ Data Protection Officer, who judged there are no data protection risk with the release of this dataset. We recognize such discussion is missing in the main paper and will surface and summarize our discussions on data collection and protection currently recorded in the accompanying datasheet. This will be done after Line 293 in the original manuscript.
>
> We also identified two potential risks in misusing the dataset and explicitly warn against such use in the original manuscript. These include:
> 1.	(Lines 291-293) The financial risk to ASOS, a multinational business, and its employees due to misrepresentation of ASOS’ overall business operations, digital product development or experimentation program operations using this dataset, which reflects the experiments run within a particular business unit; and
> 2.	(Lines 324-328) The technical risk where a (statistically) biased model/procedure is developed due to the use of this as a full performance benchmark, in other words overfitting to this dataset. The problem is pronounced for OCEs (as opposed to a generic ML task) as real online controlled experiments generally do not have a notion of ground truth.
>
> Having that said, we appreciate there is ample scope to flesh out the potential negative societal impact of releasing the dataset and how we can mitigate against such risks. We will include a discussion on two more ethical implications of releasing the dataset and rearrange the text so that the discussion on the four risks occurs together after Line 293:
>
> **Generalizing learnings to other application domains and demographics**: An OCE dataset, no matter how anonymized it is, reflects the behavior of its participants under a certain application domain. We urge potential users of this dataset to exercise caution when attempting to generalize the learnings---our assessment is that statistical methods being applied/demonstrated on this dataset are generalizable to other application domains and demographics, but the learnings themselves (e.g. the value of the hyperparameter learnt using this dataset) are not.
>
> **Running online controlled experiments in general**: The dataset is released with the aim to support experiments with adaptive stopping, which will enable faster experimentation cycle. As we run more experiments, the ethical concerns for running OCEs in general will naturally mount, as they are ultimately human subjects research. We reiterate the importance of the following three principles when we design and run experiments [1,2]: respect for persons, beneficence (properly assess and balance the risks and benefits), and justice (ensure participants are not exploited), and refer readers to Chapter 9 of [3] and its references for further discussions in this area.
>
> ------
> [1] National Commission for the Protection of Human Subjects of Biomedical and Behavioral Research. The Belmont Report: Ethical Principles and Guidelines for the Protection of Human Subjects of Research. [Bethesda, Md.]: The Commission, 1978.
> [2] Office for Human Research Protections. Federal Policy for the Protection of Human Subjects (‘Common Rule’), 2018. URL https://www.hhs.gov/ohrp/regulations-and-policy/regulations/common-rule/index.html
> [3] Ron Kohavi, Diane Tang, and Ya Xu. Trustworthy Online Controlled Experiments: A Practical Guide to A/B Testing. Cambridge University Press, 1st Edition, 2020. ISBN 9781108724265.

---

> ### Author Response · Authors · 2021-09-28
> **Response to Reviewer rsHD (part 2)**
>
> > The authors did not include […] the limitations of the study for generalizability.
>
> We believe the data dimensions defined in the taxonomy (Section 2) is generalizable to all application domains (be it agricultural, medical, or political) that run randomized controlled experiments (more pedantically, those that aim to validate/measure the average treatment effect using statistical tests). Moreover, the mapping between the taxonomy and statistical tests (Section 4) can be applied to any scenario that involves a two-sample statistical test.
>
> We will add a sentence describing the generalizability of the taxonomy and the mapping in the Introduction / around Line 89, when we introduce the mapping in Section 4 / around Line 173, and after Line 334 in the conclusion.
>
> The survey (Section 3) has its scope limited to online controlled experiments, with ample opportunity to be expanded to also include datasets for uplift modeling, quasi-experiments, and observational studies. We will expand the second half of the sentence in Line 342-344 to also mention uplift modeling and observational studies which associated datasets will be included as future work.
>
>
>
> We hope the proposed revisions have addressed your concerns, particularly on the ethical implication front. Despite our best efforts, we are aware you might have had some specific ethical concerns in mind which we might still not have addressed, if that is the case we would appreciate if you can point those out so we can consider them in detail.

---

### Official Review · Reviewer_mypc · 2021-09-21
**Paper Review**

**Rating:** 7
**Confidence:** 3

**Strengths:**

- tackling the important problem of OCE, which will be of great relevance to the industrial applications of machine learning

- first rigorous survey on existing for OCE datasets

- release of a unique dataset related to OCE, which solid documentations


**Weaknesses:**

- maybe there are other related fields and datasets, such as the Criteo dataset in the uplift modeling literature: https://ailab.criteo.com/criteo-uplift-prediction-dataset/

- the usage of the dataset is a bit unclear, the authors state that “We do caution readers on the risk using this dataset as a full performance benchmark—unlike in many machine learning tasks, real online controlled experiments generally do not have a notion of ground truth.” I basically agree, but it would be great to list some potential use cases of the dataset more specifically (and also relevant initial experiments). It would help the potential users who want to do research in the related field using the dataset.


**Additional Feedback:**

For suggestions for improvement, please see the “Weaknesses” section.

**Clarity:**

The writing is very clear and easy to follow. The relevant methods are clearly introduced with some math notations.

**Correctness:**

The procedures of survey and demonstrations seem to be well-designed by the authors

**Documentation:**

The documentation in the supplementary material looks enough.


**Ethics:**

Does not apply

**Relation To Prior Work:**

The related resources are rigorously summarized in Section 3.

**Summary And Contributions:**

The paper argues that the scarcity of public datasets and the lack of a systematic review and categorization hinders the development of OCE. Then, it presents the first survey and taxonomy for OCE datasets. The survey highlights the lack of a public dataset to support the design and running of experiments with adaptive stopping. The authors then release the first such dataset, containing daily checkpoints of decision metrics from multiple, real experiments run on a global e-commerce platform. Finally, the authors demonstrate how to use the dataset in the adaptive stopping scenario.

---

> ### Author Response · Authors · 2021-09-28
> **Response to Reviewer mypc (part 1)**
>
> Thank you very much for your review and valuable suggestions for improvement – we are delighted to hear that you agree with our assessment that OCE is of great relevance to the industrial applications of ML and the dataset is unique and hence valuable.
>
> > maybe there are other related fields and datasets, such as the Criteo dataset in the uplift modeling literature: https://ailab.criteo.com/criteo-uplift-prediction-dataset/
>
> We deeply appreciate the lead on the uplift modeling literature. Indeed, uplift modeling tasks often start with running a randomized controlled trial online, which makes any dataset arising from such tasks technically a dataset describing an online controlled experiment.
>
> We decided not to directly include the datasets related to the uplift modeling and individual treatment effect estimation tasks in Table 1 due to the drastic difference in the nature of the tasks. Many online controlled experiments concerns validating/measuring the average treatment effect (ATE) across the population using a statistical test, whereas uplift modeling concerns modelling the conditional average treatment effect (CATE) for each individual user using some form of modeling (e.g. regression) making it related to general causal inference which is out of scope.
>
> Moreover, The uplift modeling datasets we found (from the [Wikipedia page]( https://en.wikipedia.org/wiki/Uplift_modelling) and search results on Google / Google Dataset Search / Kaggle using the keywords “uplift modeling”, “individual treatment effect”, and “incrementality test”) are also generally response-granular (on randomization unit level) but not time-granular, and hence are not prime candidates for running experiments with adaptive stopping.
>
> Still, we agree the work will be further strengthened by including the discussion above, as it helps positioning our contribution more clearly. We will include both:
> - The Criteo Uplift Modeling Dataset, ideally the v2 dataset described in [this paper](https://openreview.net/pdf?id=osfJ3Ac9j9J) (under review in the same conference), and
> - The [Hillstrom / MineThatData E-Mail Analytics And Data Mining Challenge Dataset](https://blog.minethatdata.com/2008/03/minethatdata-e-mail-analytics-and-data.html) cited in the Criteo paper
>
> and the discussion above as part of the related work after Line 152, and mention including the uplift modeling as the next obvious step in generalizing the survey (together with quasi-experiments and observational studies) in Line 344.
>
> We will also include further references to datasets describing experiments in the medical and political fields to complement the existing example on agriculture experiments (Line 151). These datasets are outside the scope of the survey as they describe offline experiments and hence are included as related work instead of directly in Table 1.

---

> ### Author Response · Authors · 2021-09-28
> **Response to Reviewer mypc (part 2)**
>
> > the usage of the dataset is a bit unclear, the authors state that “We do caution readers on the risk using this dataset as a full performance benchmark—unlike in many machine learning tasks, real online controlled experiments generally do not have a notion of ground truth.” I basically agree, but it would be great to list some potential use cases of the dataset more specifically (and also relevant initial experiments). It would help the potential users who want to do research in the related field using the dataset.
>
> We agree there is scope to better articulate the use case, and will reorganize the “experiments on the dataset” section (Lines 294-328) to demonstrate the dataset can be used to support methods in the following areas:
> 1.	Meta-analyses
> 2.	Design and running of experiments with adaptive stopping
> 3.	Quasi-benchmark to compare adaptive stopping methods and their hyperparameters
> 4.	(Estimator) bias detection across time
> 5.	Learning the correlation structure across experiments, metrics, variants and time
>
> We believe items 1 and 2 are already adequately covered by the initial experiments and references featured in Lines 298-303 and Lines 304-318 in the manuscript.
>
> For item 3, we will include a new investigation on the dataset after Line 318. While real OCEs generally do not have a notion of ground truth, we concede (and hope you would also agree) that it is acceptable to use a quasi-ground truth to enable comparison between two hyperparameter settings of the same adaptive stopping method or two adaptive stopping methods.  In this investigation, we set, for all experiments featured in the dataset, the significant / not significant verdict from a Welch’s $t$-test at the end of the experiment as the “ground truth”, by virtue of it being the most used statistical test.
>
> We then compare this “ground truth” to the significant / not significant verdict of a mixed Sequential Probability Ratio test (mSPRT) at different stages of individual experiments. This yields many “binary confusion matrices” over different stages of individual experiments where a “Type I error” corresponds to cases where a Welch’s t-test gives a not significant result and a mSRPT reports a significant result. This is possible as we did not perform early stopping on any of the experiments featured in the dataset and allowed the said experiments to run for the full designed duration. This setup allows us to perform sensitivity analysis and optimization on the hyperparameters of mSPRT under what can be construed as “precision-recall” tradeoff of statistically significant treatments.
>
> It may not be feasible to run and describe the experiments for items 4 and 5 due to page constraints. Instead, we will briefly describe what examples are there and point reader to the related work (e.g., [1], where the authors describe a method to detect novelty effect---where a treatment starts strong yet falters as the newness effect recedes).
>
> [1] Nanyu Chen, Min Liu, and Ya Xu. 2019. How A/B Tests Could Go Wrong: Automatic Diagnosis of Invalid Online Experiments. In _Proceedings of the Twelfth ACM International Conference on Web Search and Data Mining (WSDM '19)_. Association for Computing Machinery, New York, NY, USA, 501–509. DOI: https://doi.org/10.1145/3289600.3291000

---

### Official Review · Reviewer_naDm · 2021-09-24
**Datasets for Online Controlled Experiments**

**Rating:** 7
**Confidence:** 3

**Strengths:**

Overall, well-motivated and clearly-written paper with very useful contributions. Survey and taxonomy seem quite useful, and dataset seems generally well-constructed.

**Weaknesses:**

- While the taxonomy they present seems adequate for running statistical tests, as they demonstrate, I would also expect to see more space dedicated to the context of the application domain and assumptions underlying the dataset's creation. They make a case for the need to understand dataset context--"We also recognize that in practice data are often used for purposes beyond what it is originally collected for"-- but don't address this fully outside of the literal statistical tests that are able to be performed. I think it would be simple but impactful to intentionally surface key details from the documentation explicitly as part of your taxonomy-- e.g. "Majority of users in North America and Europe, English-language, unknown or undocumented target demographics)". This could also address issues related to the fairness analysis of OCE data (e.g. Was the website screen reader enabled?).

While this doesn't strictly have to be added in v1, I think it would be helpful to address this gap in your discussion.

**Additional Feedback:**

See weaknesses.

**Clarity:**

Yes, very much so. (Tiiiiny nit: inconsistent capitalization of "timestamp" in Table 1?)

**Correctness:**

Design and taxonomy design both seem fine; not qualified to assess low-level statistical details.

**Documentation:**

Documentation looks good.

**Ethics:**

See "weaknesses"

**Relation To Prior Work:**

Well documented in sections 2 and 3; I'm not familiar with any major work they are missing.

**Summary And Contributions:**

This paper highlights that a lack of publicly available OCE datasets makes it difficult to advance work in this space. To this end, they present a survey of existing datasets along with a taxonomy for classifying OCE data. They also present their own OCE dataset, one which allows for experiments that allow adaptive stopping, a key feature that is both central to current decision-making practices in OCEs and missing from existing datasets. They demonstrate how this dataset can be appropriately used as a "common platform" for running a wide variety of statistical tests.

---

> ### Author Response · Authors · 2021-09-28
> **Response to Reviewer naDm**
>
> Thank you for your favorable review and suggestions on further expanding the taxonomy to ensure one understands the dataset context.
>
> > I would also expect to see more space dedicated to the context of the application domain and assumptions underlying the dataset's creation. They make a case for the need to understand dataset context […] but don't address this fully outside of the literal statistical tests that are able to be performed.
>
> As you have pointed out, the original manuscript contains a column describing the application domain of the dataset, though we agree that it is important to expand on and discuss the reasons why we many want to differentiate on the domain, the most obvious one being the learnings from a dataset from a certain domain may not translate to another domain. We will expand the paragraph in Lines 130-134 to include the above.
>
> > I think it would be simple but impactful to intentionally surface key details from the documentation explicitly as part of your taxonomy-- e.g. "Majority of users in North America and Europe, English-language, unknown or undocumented target demographics)". This could also address issues related to the fairness analysis of OCE data (e.g. Was the website screen reader enabled?)
>
> We did not surface any temporal, geographical and demographic information about the experiment(s) as we believed at this stage there are not enough non-nulls for such information to be useful. Indeed, we are still unable to identify the target demographic for the bottom seven datasets in Table 1, though two of them are synthetic and thus should not have one. We do appreciate that surfacing such information where available is a relatively simple way to provide more dataset context and hence strengthen the survey.
>
> To the best of our knowledge, the target demographic for the other five datasets is as follow:
> * Upworthy: Mostly English-speaking users in USA
> * ASSISTments: Mostly US middle school students (age 11-14) in/near Massachusetts, USA
> * Library (A/B Testing Web Analytics Data): English-speaking university library users
> * MeuTutor: Brazilian high school students taking ENEM (age 17)
> * Code Review: Mostly Dutch/Swiss software developers & graduate students
>
> We will update Table 1 to include the above information, together with when the experiments are run (again, to the best of our knowledge). We will also include a couple sentences after Line 134 to cover the inclusion of the two new attributes and their significance, including the example you have suggested on fairness analysis.
>
> Aside: The above discussion has prompted us to review whether we have provided sufficient information on the target demographics. ASOS’ target demographic is mainly 20-somethings based in North America, Europe and Oceania, where the company’s key markets are located. We will include this information in our datasheet.
>
> > (Tiiiiny nit: inconsistent capitalization of "timestamp" in Table 1?)
>
> We will fix the capitalization for “timestamp” in Table 1 - the “timestamp” in the third data entry should indeed be in lower case.

---

### Official Review · Reviewer_BKGH · 2021-09-25
**Review of "Datasets for Online Controlled Experiments"**

**Rating:** 6
**Confidence:** 1
**Correctness:** The construction of the dataset is so…
**Clarity:** The paper is well-written.

**Strengths:**

This work is significant as the first to provide a dataset for investigating early stopping methods in OCE, and should help the development of these methods. Also significant is the introduction of a taxonomy for OCE datasets and the categorization of existing datasets within the proposed taxonomy.

**Weaknesses:**

One weakness of this dataset is that it may not be very relevant to the broader machine learning community.

**Additional Feedback:**

No additional feedback.

**Documentation:**

There is sufficient information about each of these areas in both the paper and datasheet.

**Ethics:**

I do not see any ethical concerns.

**Relation To Prior Work:**

The limitations of previous work for testing adaptive stopping methods are clearly discussed, as well as the advantages of the proposed dataset over these methods.

**Summary And Contributions:**

The main contribution of the paper is to introduce a novel dataset that is the first to allow the testing of adaptive stopping methods for Online Controlled Experiments (OCE). The paper is also the first to survey existing public OCE datasets, and categorizes them according to four taxonomic dimensions. Also included is a discussion of some common statistical tests and when they are applicable.

---

> ### Author Response · Authors · 2021-09-28
> **Response to Reviewer BKGH**
>
> Thank you very much for your review. We would like to address your concern on the relevancy of the proposed dataset.
>
> > Weaknesses: One weakness of this dataset is that it may not be very relevant to the broader machine learning community.
>
> The ASOS Digital Experiments Dataset was indeed collected and prepared with a fairly narrow purpose in mind - to support the design and running of experiments with adaptive stopping, though we argue the broader machine learning community will benefit from having this dataset publicized. Online controlled experiments are now an essential step within the machine learning development lifecycle in the industry, where one validates/measures the real impact of a proposed ML model. Having early stopping methods means we will be able to speed up the experiment lifecycle. This enables a ML model to be evaluated more quickly and more ML models to be evaluated. We will, in our revision, articulate the fact that OCE is an essential step in the industry ML development cycle in Line 20.
>
> Moreover, the multi-experiment, multi-time series nature of this dataset also supports more general applications in the ML field. We believe the dataset can be used to support methods in the following areas:
> 1. Meta-analyses
> 2. Design and running of experiments with adaptive stopping
> 3. Quasi-benchmark to compare different adaptive stopping methods
> 4. Bias (of the estimator) detection across time
> 5. Learning the correlation structure across experiments, metrics, variants and time
>
> We will reorganize the “experiments on the dataset” section (Lines 294-328) to better articulate the use cases, describe relevant experiments, and provide references that describe relevant methods.
>
> Finally, we believe the ML community will also benefit from the systematic mapping between common statistical tests and their data requirements, another main contribution in our work. The mapping can be applied to any scenario that involves a two-sample statistical test, far beyond the scope of online controlled experiments. These include offline experiments (e.g. comparing the performance between different ML models) and experiments without a randomized control (e.g. counterfactual learning / causal ML). We believe almost every ML researcher/practitioner will encounter the former at some point in their career, whilst the latter is the focus for many in the ML community right now.
>
> Having that said, we realize we had not made the generalizability of the mapping beyond online controlled experiments clear in our submission, and will emphasize so in the updated manuscript (in the Introduction / around Line 89, when we introduce the mapping in Section 4 / around Line 173, and after Line 334 in the conclusion).

---

### Author Response · Authors · 2021-09-30
**We will upload our revised manuscript by 30 Sep AoE**

Thank you for your reviews and valuable suggestions to further improve the work. We have responded to each review individually with our thoughts and proposed edits, and hope they have sufficiently addressed any concerns that you might have.

We agreed with the chairs that we should submit our revisions by 30 Sep AoE (24 hours after the normal deadline), and will upload our revised manuscripts with the proposed edits by then. We are, of course, more than happy to take further comments and suggestions between then.

---

### Author Response · Authors · 2021-10-01
**Revised manuscript - summary of changes**

We thank the reviewers once again for their valuable comments and suggestions. We have addressed all the points raised by them and applied the changes to the manuscript according to our previous replies.

See below for a detailed list of the changes.  The line numbers quoted in this comment refers to the PDF modified on 01 Oct 2021, whereas the line numbers in our responses to individual reviewers refer to the previous PDF dated 28 Aug 2021. The identifiers in the round brackets corresponds to the reviewer whose comments/suggestions the changes intend to address.

**Multiple Sections (1, 2, 4, 6)**
* Emphasized the generalizability of the taxonomy and mapping as applicable to all experiments (e.g. offline, non-randomized controlled) in Lines 88, 131-132, 189-190, and 394 (BKGH, rsHD)

**Section 1 - Introduction**
* Articulated the fact that OCE is an essential step in industry ML development life cycle in Line 20 (BKGH)

**Section 2 - Taxonomy**
In Lines 134-141:
* Expanded the paragraph on application domain and discuss the motivation for inclusion (naDm)
* Included motivation of including target demographics and temporal coverage (naDm)

**Section 3 - Survey**
* Fixed inconsistent capitalization of "timestamp" in Table 1 (naDm)
* Added missing references to the 2 Udacity course datasets and blog post / webpage documentations in Table 1
* Included the fact that the target demographics for ASOS is "20-somethings based in North America, Europe, and Oceania." in the datasheet (naDm)
* Included references to and a discussion on uplift modeling literature in Line 164 (mypc)
* Included more references to experiment datasets for randomized controlled trials run offline in Line 162 (mypc)
* To include: target demographics and temporal coverage information in Table 1 (naDm) - we have included the data in the response to the reviewer, and will update and reformat the table in the camera-ready version

**Section 5 - Dataset**
* Reorganized the "experiments on the dataset section" to better articulate use cases, describe relevant experiments, and provide references that describe relevant methods as Section 5.1 (BKGH and mypc)
* Added experiment on a quasi-benchmark to compare adaptive stopping methods and their hyperparameters in Lines 332-343 (mypc)
* Motivated and provided references for (1) bias (of the estimator) detection across time and (2) learning the correlation structure across experiments, metrics, variants and time in Lines 344-348 (mypc)
* Surfaced discussion on data protection in the accompanying datasheet in Lines 352-357 (rsHD)
* Reorganized potential misuses of the dataset and include two more ethical implications of releasing the dataset in Lines 358-378 (rsHD)

**Section 6 - Conclusion**
* Mentioned the possibility to generalizing the survey to cover uplift modeling, together with quasi-experiments and observational studies, as future work in Lines 392-393 (mypc, rsHD)

---

### Decision · Program_Chairs · 2021-10-09

**Decision:**

Accept

**Comment:**

Reviewers all agree that this paper offers a valuable contribution to this track. While the confidence of the reviewers ratings is lower than I would have desired, the two highest scoring reviews (7/7) are the highest confidence (3). The authors engaged with the constructive reviews, responding to many of the comments and addressing most of the weaknesses. The authors have also revised their manuscript in light of the reviews and discussion.

Given the positive reviews and in-depth author response, I am recommending acceptance.